# Sustainable Diet Dimensions. Comparing Consumer Preference for Nutrition, Environmental and Social Responsibility Food Labelling: A Systematic Review

**Rebecca C. A. Tobi** [1],*, **Francesca Harris** [2], **Ritu Rana** [3], **Kerry A. Brown** [4],*, **Matthew Quaife** [4] **and Rosemary Green** [2]

1   Department of Population Health, London School of Hygiene & Tropical Medicine, London WC1E 7HT, UK
2   Department of Population Health, LSHTM Centre on Climate Change and Planetary Health, London School of Hygiene & Tropical Medicine, Keppel St, London WC1E 7HT, UK; Francesca.Harris@lshtm.ac.uk (F.H.); Rosemary.Green@lshtm.ac.uk (R.G.)
3   Indian Institute of Public Health Gandhinagar, Gujarat 382042, India; rrana@iiphg.org
4   Faculty of Public Health & Policy, London School of Hygiene & Tropical Medicine, Tavistock Place, London WC1H 9SH, UK; matthew.quaife@lshtm.ac.uk
*   Correspondence: rca.tobi@gmail.com (R.C.A.T.); Kerry.Brown@lshtm.ac.uk (K.A.B.); Tel.: +44-207-927-2112 (K.A.B.)

**Abstract:** Global food systems are currently challenged by unsustainable and unhealthy consumption and production practices. Food labelling provides information on key characteristics of food items, thereby potentially driving more sustainable food choices or demands. This review explores how consumers value three different elements of sustainable diets: Comparing consumer response to nutrition information on food labels against environmental and/or social responsibility information. Six databases were systematically searched for studies examining consumer choice/preference/evaluation of nutrition against environmental and/or social responsibility attributes on food labels. Studies were quality assessed against domain-based criteria and reported using PRISMA guidelines. Thirty articles with 19,040 participants met inclusion criteria. Study quality was mixed, with samples biased towards highly-educated females. Environmental and social responsibility attributes were preferred to nutrition attributes in 17 studies (11 environmental and six social), compared to nine where nutrition attributes were valued more highly. Three studies found a combination of attributes were valued more highly than either attribute in isolation. One study found no significant preference. The most preferred attribute was organic labelling, with a health inference likely. Consumers generally have a positive view of environmental and social responsibility food labelling schemes. Combination labelling has potential, with a mix of sustainable diet attributes appearing well-received.

**Keywords:** food labelling; sustainable diet; ecolabels; nutrition labels; social responsibility labels; organic labelling; animal welfare labels

## 1. Introduction

Globalisation and urbanisation are driving a shift towards unhealthy dietary patterns associated with rising rates of nutrition-related chronic diseases [1,2]. Meanwhile, agriculture is a major contributor to climate change, with food systems responsible for around 24% of global anthropogenic greenhouse gas emissions [3]. Simultaneously, climate and other environmental changes are posing new threats to food production [4]. In order to improve both human and planetary health a move towards more sustainable food systems is therefore required. Sustainable diets are food consumption patterns that are

beneficial for human health, nutrition, environmental, ethical and economic domains [5–7]. Such diets have been gaining an increasing amount of attention from researchers and policy-makers and are gradually becoming better defined [8]. However, questions remain regarding public acceptability of such dietary patterns, and how best to encourage a shift towards more sustainable and healthy diets. Food labelling may be one tool with which to encourage consumer uptake of such diets. Yet there is some uncertainty as to the efficacy and acceptability of both nutrition and 'eco' labels, with both governments and manufacturers reluctant to entirely embrace mandatory or uniform labelling schemes.

Food labelling is increasingly used globally, with a number of different nutrition, environmental and social responsibility labelling schemes in existence. Nutrition food labelling schemes are widely used in high-income countries, with an estimated 48% of European foods carrying voluntary front-of-pack labels [9]. Messaging is diverse, ranging from standard macronutrient content information to health claims, with both mandatory and voluntary schemes co-existing. In Europe the increasing use of labelling schemes has led to formal regulatory controls for both mandatory and voluntary schemes, with Regulation (EC) 1924/2006 of the European Parliament and Council harmonizing regulation of nutrition and health claims on food packaging, Regulation (EC) 1169/2011 defining the provision of food information to the public and Regulation (EC) 66/2010 describing use of the voluntary EU ecolabel scheme. The available evidence shows an association between the use of nutrition labels and healthier diets [10–13], although there is a lack of high-quality studies [10–12]. Environmental and social responsibility labels in contrast are mostly voluntary or privately-run schemes, with the Global Ecolabelling Network listing 463 different schemes—over 120 of which are active in the EU [14,15].

Recent years have seen notable industry engagement with sustainability labelling, with 2008–12 producing an average annual growth rate of 50% for 16 of the main sustainability schemes covering the ten most labelled agro-food products [16]. Consumer acceptance of such schemes has been mixed, with Tesco's carbon-labelling scheme discontinued in 2012 due to insufficient demand [17], and with continuing accusations of corporate 'green-washing' or of over-exaggerating sustainability claims to gain market share [16,18]. Environmental and social responsibility labels are therefore often considered less impactful than other product information, with one study concluding that sustainability labels do not play a major role in food choice, reporting low levels of use [19]. Despite these findings, meta-analyses looking specifically at socially responsible products have indicated consumers are willing to pay a premium [20,21]. This theoretically should encourage wider use of such schemes, but there is little consensus on how best to profile sustainable consumers [22]. Overall, research is limited; the most recent literature review of ecolabels found dates back to 2002 [23].

While labelling initiatives separately considering nutrition, environmental and social issues have existed for several decades, few systems have attempted to combine multiple product attributes despite the overlap between more healthful and sustainable diets [24,25]. With increasing interest in such schemes, it is important to understand their effect on consumer decision-making before they become widely used. For example, the use of multiple or integrated labelling schemes may result in confusion for the consumer and ultimately deter purchasing. Integrated labelling may also lead to competing effects between different attributes. While labelling schemes have been reviewed individually in the literature, no previous reviews on the integrated approach have been identified.

The aim of this study was to explore the effect of different labelling schemes associated with three elements of sustainable diets on consumer choice: nutrition, environment and social responsibility. We systematically reviewed studies testing consumer choice and preference for each 'attribute', defined as a characteristic of a product that impacts on the consumer's purchasing decision. We review studies that specifically compare nutrition to the other attributes as nutrition information is currently the more widely used and regulated type of food labelling scheme. We compare the preference for each attribute and explore the characteristics of consumers in order to inform future research and interventions. Our findings indicate that environmental and social responsibility food labelling schemes are of more value to consumers than previously thought, and preferred when in direct comparison to nutrition labels. However, an overlap in consumer perceptions of 'good for health' and 'good for the

planet' suggest care must be taken to prevent consumers drawing unmerited health inferences from environmental labels.

## 2. Materials and Methods

This systematic review follows the PRISMA Checklist (preferred reporting items for systematic review and meta-analysis protocols). Six databases were searched covering a number of specialty areas in an attempt to ensure an interdisciplinary search strategy. This included medical and health sciences (Global Health, Web of Science and PubMed), environmental science (GreenFILE), business and management (ABI Inform) and psychology (PsychINFO).

A search strategy was developed and subsequently adapted for each database. No limitations on publication date were set. Search terms included the concepts; "sustainability", "label", "food" and "consumer", with a second sub-search combining "eco-label" with "food" and "consumer" (Table 1). Given the large number of different labelling schemes, a horizontal approach was adopted for search terms by focussing on broad product attributes (e.g., animal welfare) instead of individual labelling schemes (e.g., RSPCA assured) [26].

**Table 1.** Search strategy.

| | **AND** | | | | |
| **OR** | **Environment and Social Responsibility** | **Label** | **Food** | **Consumer** | **Sub-Search** |
| --- | --- | --- | --- | --- | --- |
| | Carbon footprint* | Label * | Food * | Consumer * | Eco-label * |
| | Water footprint * | Claim * | Nutrit * | Shopper * | Ecolabel * |
| | Environment * | Packet * | Diet * | Buyer * | Carbon-label * |
| | Voluntary sustainability standard * | Packag * | Sustainable diet * | Decision-making | Ethical label* |
| | Low-carbon | Label?ing | Health claim * | Point-of-purchase | |
| | Organic | Certificat * | | Purchas * decision * | |
| | Biodiversity | Awareness | | Choice * | |
| | Green label * | Standard * | | Willingness-to-pay | |
| | Green | Signal * | | | |
| | Palm oil free | | | | |
| | Environment * protection | | | | |
| | Conservation | | | | |
| | Social * responsibl * | | | | |
| | Social * | | | | |
| | Sustainab * | | | | |
| | Fair trade | | | | |
| | Animal welfare | | | | |
| | Marine stewardship | | | | |

Note: * served as the truncation (or wildcard) operator using Boolean Search Operators.

Search strategies can be found in the additional information (Supplementary Information 1). Database searches were completed by 5 January 2019. Study screening and selection was carried out using Mendeley v1.19.

### 2.1. Study Selection

Inclusion criteria were predefined to ensure study selection relevance, with eligibility criteria formatted following the population, intervention, comparison, outcomes, situation, type of study (PICOST) framework [27]:

- Population—independent consumers and/or purchasers of packaged foods aged 18–75 years old.
- Intervention—labelling/logos/claims/information relating to nutritional, environmental and/or social responsibility product attributes designed for display on packaged foods.
- Comparison—consumer preference for nutrition attributes were compared to environmental and/or social responsibility attributes on food labels.

- Outcome—qualitative outcomes included consumer evaluation, interpretation and liking of different attributes. Empirical outcomes included attribute utility estimates and willingness-to-pay.
- Situation—no geographical limits.
- Type of study—primary research studies only.

Three attribute areas were selected as a focus for the study, with databases selected to recognise food choice research designs used in different disciplines. Sustainable food system frameworks (Figure 1) often include economic and cultural domains in addition to health, social and environmental ones. However, these were excluded from this study in line with other literature on the topic viewing such schemes (e.g., protected designated origin labels) as markers first and foremost of geographical indication, more reflective of ethnocentric concerns than altruistic or environmental ones [19,28,29].

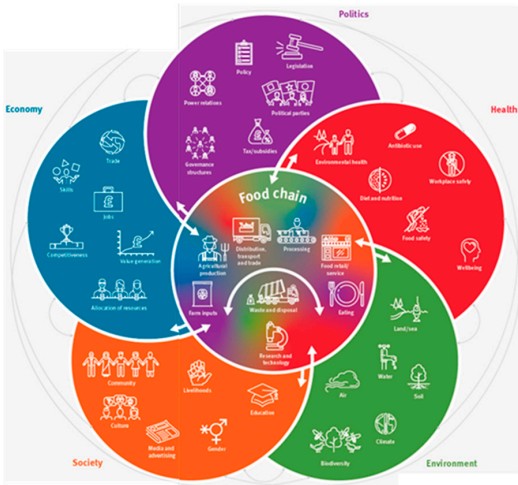

**Figure 1.** Domains within a sustainable food system framework [30].

The nutrition attributes included in this review related to the nutritional content and nutrition-health claims of food. This included labelling schemes displaying information related to food energy and nutrient content (macro and micro), nutrition-related health claims, and the 5-a-day logo. Sensory and hedonic attributes such as flavour and textural preferences were excluded.

Environmental attributes included labelling schemes related to carbon emissions, water footprints, more general ecological and environmental sustainability claims, biodiversity and organic production methods. Schemes with an insufficient evidence-base regarding their environmental impact, e.g., genetically modified organisms [31], were excluded.

Social responsibility attributes were included when they related to human or animal welfare or equity, notably fair trade and animal welfare labels. Remaining exclusion criteria can be seen in Box 1.

Titles and abstracts of studies were first screened for relevance, before remaining studies were screened using a full-text article review form. A 33% sample was screened in duplicate to minimise potential bias.

**Box 1.** Exclusion criteria.

➢ Non-English language papers
➢ Not peer-reviewed
➢ Studies testing non-food products; alcohol, supplements, coffee
➢ Interventions using a place (e.g., canteens) not a food product to display attribute information
➢ Study populations comprised of retailers or manufacturers
➢ Products and/or study populations biased towards one of the three attribute domains e.g., organic consumers only

### 2.2. Data Extraction and Quality Assessment

Data were extracted from eligible studies, including: Bibliographic information, study setting and sample size, sample socio-demographic characteristics, study design, behaviour change models used, food product(s) and relevant attributes tested (nutrition, environmental and/or social). Relevant outcomes—mainly estimates of attribute utility—were also extracted, in addition to any details provided on study funding source and conflict of interest (COI) declarations. Data extraction was validated on 38% of the sample by a second reviewer.

In line with Cochrane recommendations [32], a domain-based evaluation was used to categorise studies (criteria met, not met, unclear). As the majority of reviewed studies were choice experiments an assessment form was designed to appropriately assess these studies. Criteria were based on the "good-practice" checklist of Lancsar and Louviere [33]; a method used in two recent systematic reviews of choice experiment literature where the criteria were also adapted to suit reviewed studies [34,35]. Two criteria relating to single dimensional attribute criteria and suitability of econometric choice model were excluded to avoid disadvantaging the alternative study designs included. To this end a criterion around response rate was included. An additional criterion assessing funding source was added given the potential for conflict-of-interest bias due to vested industry interest in food labelling schemes. Further explanation of the criteria used in this review's quality assessment can be found in the additional information (Supplementary Information 2).

A narrative approach was adopted to explore included studies as the heterogeneity in study designs and outcomes prevented a quantitative analysis. The designs of included studies were choice experiments, experimental auctions and ranked choice/preference surveys. These are hypothetical study designs but offer insight into purchasing 'trade-offs' which are not always present in observational studies. Choice experiments are based on theories of economic rationality [36], with relative attribute utility and willingness-to-pay (WTP) the main outcomes measured. Experimental auctions use consecutive bidding rounds to determine consumer attribute preference and thus aim to reduce hypothetical bias, with WTP the primary outcome analysed. For those studies asking participants to rank their label preferences, label liking and willingness-to-buy were relevant outcomes for analysis. One pre-post randomised controlled experiment was included, comparing mean post-affect scores subsequent to the messaging intervention.

### 2.3. Analysis

The principal summary measures analysed were estimates of attribute utility, attribute or label rank order and WTP. We ranked consumer preferences for the nutrition, social responsibility and/or environmental attributes tested according to relative magnitude within studies depending on whether attributes were chosen more frequently, ascribed a higher utility or rank, or evaluated more positively by consumers. These findings were then compared between studies. The different labels within each of the three attribute areas under investigation were also analysed and mapped against the food products tested for each individual study. Where WTP as a percentage premium is provided or can be calculated these results have been reported given WTP's value in estimating product attribute competitiveness. A meta-analysis was not conducted given the nature of choice experiment results (where coefficients differ in scale between experiments and populations) and study heterogeneity.

## 3. Results

### 3.1. Study Selection

After completing a two-stage screening process, 30 studies remained that met eligibility criteria (Figure 2).

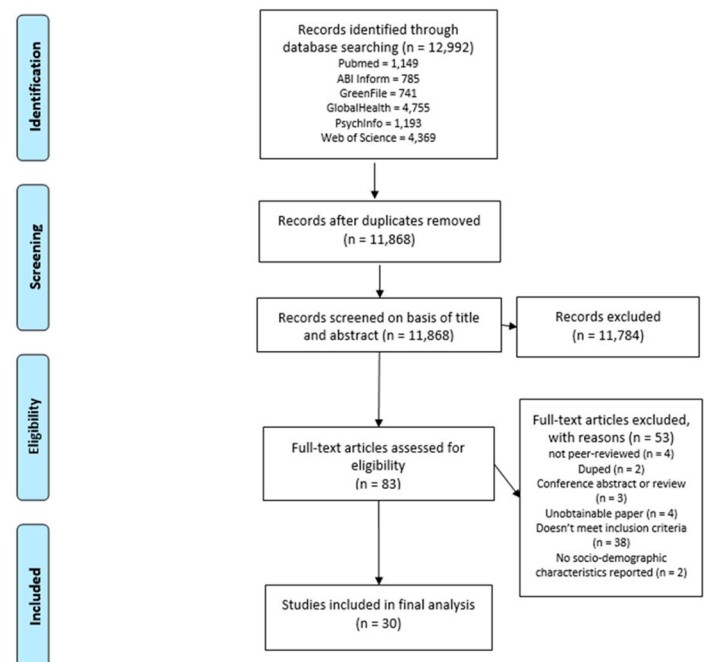

**Figure 2.** Preferred reporting items for systematic reviews and meta-analyses (PRISMA) flow diagram showing study selection [37].

*3.2. Study Quality*

Based on whether quality criteria were met, not met or inadequately reported, studies were categorised as high, medium or low quality (Figure 3). Studies were broadly of sufficient quality, with ten classified as high quality, eight as medium and twelve as low. Response rates were poorly reported despite the majority of studies using survey methodology, leaving many at risk of selection bias. While there are no universally applicable standards for acceptable survey response rates [38] only four studies clearly noted response rates of ≥50%. Thirteen studies used unrepresentative sampling frames (e.g., university populations), and seven were at risk of reporting bias with funding from the food industry. Attribute choice and experimental design descriptions were broadly carried out to a high standard, grounded in prior qualitative work.

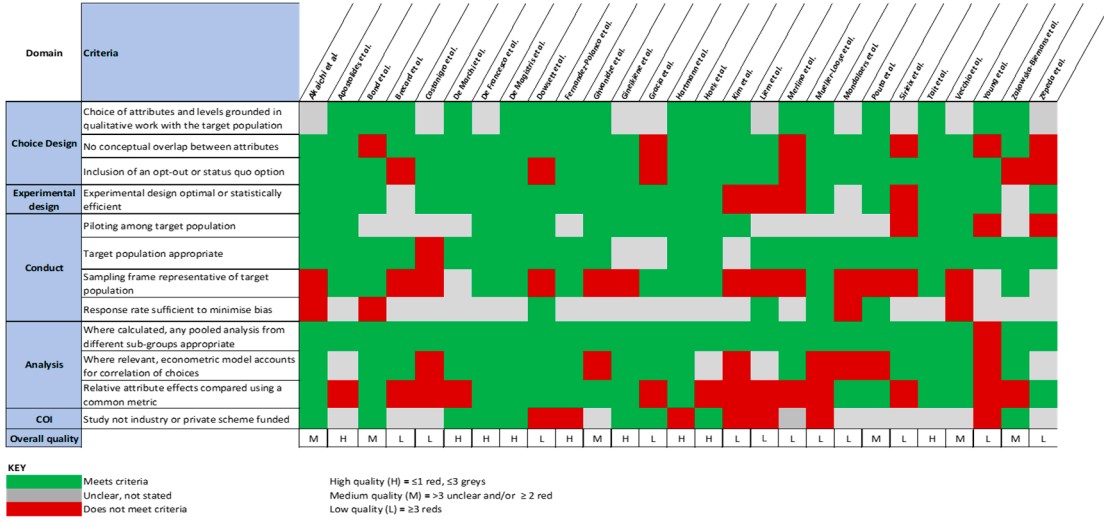

**Figure 3.** Quality assessment of reviewed studies.

### 3.3. Study Characteristics

Of the 30 papers included in the final review, all were set in high-income countries according to the World Bank classification system [39]. Of the 16 different countries investigated, the majority of studies were carried out in Europe (n = 26), followed by North America (n = 8) and Australia (n = 4). Only one study was carried out in Asia. Five papers undertook cross-country comparison. All studies were published after 2008, with 19 studies (61.3%) published in the last three years (since 2016). Measurement of outcomes varied and included attribute utility estimates (n = 11), rank ordered or best-worst scoring (n = 5), and WTP (n = 12).

The majority of studies used choice experiment designs (n = 20): A quantitative way of eliciting individual preferences by providing respondents with choice sets of hypothetical alternatives, and frequently used in food labelling studies. The remainder were experimental auctions (n = 5), preference surveys (n = 4), and one pre-post randomised controlled design. Three studies included qualitative information via focus groups [40–42]. Many studies employed Likert scales to gauge consumer preference and characteristics. Only seven studies used conceptual theories of consumer behaviour to inform their findings [42–48]. Sample size varied between studies, ranging from 16 participants to 1950, with a mean study size of 635 participants and an overall population of 19,040 participants across the 30 studies. The socio-demographic and economic characteristics of study participants revealed a bias towards females and graduates, with ten studies using samples where over 60% of participants were female. Twelve studies had samples where over 40% of participants had tertiary-level education.

### 3.4. Synthesis of Results

#### 3.4.1. Attribute Preference

Overall, 17 studies (57%) found environmental and social responsibility attributes were preferred (chosen more frequently, ascribed a higher utility or rank or evaluated more positively) to nutrition attributes by consumers. Nine studies showed a preference for nutrition or nutrition-health label information; one study finding no significant difference between descriptive labels but with a trend toward the nutrition-health label. Three studies found a mix of both health and environmental and/or social responsibility attributes were most popular [47,49,50]. In those studies where environmental and social responsibility attributes were preferred, organic was the preferred choice in eight studies [28,43,46,51–54], with animal welfare claims coming a close second (five studies) [44,55–58]. Two studies found environmental impact labels were valued most highly [42,59], one Fairtrade [40] and one carbon footprint [60]. Fourteen of the studies found that overall, consumers valued other attributes being tested simultaneously (such as price) more highly than the attributes examined here.

Of the nine studies where consumers preferred nutrition-related attributes to the environmental or social alternative, there was a split between a preference for macronutrient information and nutrition-related health claims (Table 2). Low-fat labelling information was ascribed a high utility by consumers, the preferred type of nutrition attribute information in four studies [41,45,61,62]. A distinction in consumer label information preference can be seen in those studies where two types of nutrition information (health claims and nutrient content information) are tested side-by-side. In the four studies permitting a comparison of health claims and nutrition content information [45,51,57,63], just one found that the nutrient content information was preferred to information emphasising longer-term health benefits [45]. This preference for health claims over nutrition content information was particularly strong in Zakowska-Biemans et al. [57], where the utility of nutrient claims simply stating the omega-3 and vitamin content of eggs was rated negatively compared to health claims expanding on the functional properties of these nutrients. So "vitamins A and E have a positive effect on the cardiovascular system" was more positively received (utility; 0.282) than the more basic "vitamins A and E" claim (utility; 0.055) [57].

**Table 2.** Summary of study characteristics and findings.

| Author | Study Design | Outcome Measurement | Setting | Sample Size | Comparative Ranking of Relevant Product Attributes | Sustainable Diet Attribute Preference |
|---|---|---|---|---|---|---|
| Apostolides et al. (2016) | Choice experiment | Attribute utility | UK | 247 | (1) Fat content<br>(2) Carbon footprint<br>(3) Organic | Nutrition |
| Young et al. (2016) | Choice experiment | Multidimensional scaling | USA | 218 | (1) Low fat/clean ingredients<br>(2) Low fat/some sugar<br>(3) Sustainability label | Nutrition |
| Kim et al. (2013) | Choice experiment | Attribute utility (zero-centred) | USA | 250 | (1) Fat content<br>(2) Sugar content<br>(3) Organic | Nutrition |
| Gracia et al. (2016) | Choice experiment | Direct ranking | Spain | 540 | (1) Nutritional fact panel<br>(2) Organic<br>(3) Animal welfare<br>(4) Food miles<br>(5) Carbon footprint | Nutrition |
| Ghvanidze et al. (2017) | Choice experiment | Attribute utility, WTP | USA, UK, Germany (DE) | 1872 | (1) Nutrition (UK/USA/DE)<br>(2) Ecological (UK/USA)<br>(3) Social responsibility (UK/US)<br>(4) Health (UK/USA/DE) | Nutrition |
| Mueller-Loose et al. (2012) | Choice experiment | Attribute utility | Australia | 1601 | (1) Health logo<br>(2) Carbon zero claim | Nutrition; health |
| De Francesco et al. (2017) | Choice experiment | WTP | Italy | 1566 | (1) Health claim<br>(2) Environmental claim<br>(3) Combination | Nutrition; health |
| Brecard et al. (2012) | Choice experiment | Rank ordered utility | France | 911 | (1) Health label<br>(2) Eco label<br>(3) Fairtrade label | Nutrition; health |

**Table 2.** *Cont.*

| Author | Study Design | Outcome Measurement | Setting | Sample Size | Comparative Ranking of Relevant Product Attributes | Sustainable Diet Attribute Preference |
|---|---|---|---|---|---|---|
| Bond et al. (2008) | Choice experiment | Attribute utility, WTP | USA | 1549 | (1) Health claim A, 'healthy diets can reduce disease risk' (2) Health claim B, 'fibre and vitamins can reduce disease risk' (3) Organic (4) 5-a-day logo (5) Vitamin C | Nutrition; health |
| Liem et al. (2018) | Preference survey | Liking and WTP | Australia | 119 | (1) Health (2) Social responsibility (3) Sustainability | No significant difference |
| Dowsett et al. (2018) | Pre-post randomised controlled experiment | Mean post-affect score | Australia | 460 | (1) Animal welfare information (2) Nutrition information | Social responsibility; animal welfare |
| Fernandez-Polanco et al. (2013) | Choice experiment | Attribute utility, WTP | Spain | 169 | (1) Animal welfare (2) Environmental information (3) Omega-3 content | Social responsibility; animal welfare |
| Zakowska-Biemans et al. (2017) | Choice experiment | Attribute utility | Poland | 935 | (1) Animal welfare (2) Organic (3) Vitamin health claim (4) Omega-3 health claim (5) Vitamin content (6) Omega-3 content | Social responsibility; animal welfare |
| Merlino et al. (2018) | Choice experiment | Best-worst mean scores | Italy | 401 | (1) Animal welfare (2) Organic (3) Nutrition | Social responsibility; animal welfare |
| Pouta et al. (2010) | Choice experiment | Attribute utility | Finland | 627 | (1) Animal welfare (2) Organic (3) Omega-3 content | Social responsibility; animal welfare |

**Table 2.** *Cont.*

| Author | Study Design | Outcome Measurement | Setting | Sample Size | Comparative Ranking of Relevant Product Attributes | Sustainable Diet Attribute Preference |
|---|---|---|---|---|---|---|
| Sirieix et al. (2013) | Preference survey | Label preference | UK | 16 | (1) Social responsibility <br> (2) Organic <br> (3) Nutrition labels <br> (4) Combination | Social responsibility; fair trade |
| Zepeda et al. (2013) | Choice experiment - qualitative | Ordinal ranking | France, Spain, Canada, USA | 375 | (1) Sustainable production <br> (2) Organic <br> (3) Health logos | Environmental |
| Hartmann et al. (2018) | Preference survey | Intention to pay a price premium | UK, France, Sweden, Poland | 1950 | (1) Environmental; palm oil free <br> (2) Health; gluten free <br> (3) Health; lactose free | Environmental |
| Tait et al. (2016) | Choice experiment | Attribute utility, WTP | UK, Japan | 1194 | (1) Carbon emissions <br> (2) Water efficiency <br> (3) Vitamin content | Environmental; carbon |
| Costanigro et al. (2015) | Choice experiment | Best-worse ranking | USA | 244 | (1) Organic <br> (2) Reduced fat | Environmental; organic |
| De Marchi et al. (2015) | Choice experiment | Attribute utility | USA | 173 | (1) Organic label <br> (2) Health claim <br> (3) Carbon trust label <br> (4) Calorie content | Environmental; organic |
| De-Magistris et al. (2016) | Experimental auction | WTP | Spain | 129 | (1) Organic <br> (2) Reduced fat | Environmental; organic |
| Gineikiene et al. (2017) | Preference survey | Willingness to buy | Lithuania | 295 | (1) Organic <br> (2) Nutrient enhanced | Environmental; organic |
| Mondelaers et al. (2009) | Choice experiment | WTP | Belgium | 529 | (1) Organic <br> (2) Vitamin A <br> (3) Ecological claim | Environmental; organic |

**Table 2.** *Cont.*

| Author | Study Design | Outcome Measurement | Setting | Sample Size | Comparative Ranking of Relevant Product Attributes | Sustainable Diet Attribute Preference |
|---|---|---|---|---|---|---|
| Vecchio et al. (2016) | Experimental auction | WTP | Italy | 100 | (1) Organic<br>(2) Nutrient enhanced | Environmental; organic |
| Almli et al. (2011) | Choice experiment | Willingness to buy | France, Norway | 239 | (1) Organic<br>(2) Omega-3 content | Environmental; organic |
| Cagalj et al. (2016) | Experimental auction | WTP | Croatia | 258 | (1) Organic<br>(2) Health claim<br>(3) Environmental claim | Environmental; organic |
| Hoek et al. (2017) | Choice experiment | Attribute utility | Australia | 944 | (1) Combination<br>(2) Health<br>(3) Environment | Combination |
| Lemken et al. (2017) | Experimental auction | WTP | Germany | 1020 | (1) Combination<br>(2) Health[1]<br>(3) Environment[1] | Combination |
| Akaichi et al. (2019) | Experimental auction | WTP | UK | 120 | (1) Combination<br>(2) Organic<br>(3) Animal welfare | Combination |

[1] Non significant result.

### 3.4.2. Label Characteristics

Studies examined a range of different labelling information within the three attribute areas (Figure 4). 'Organic' was the most frequently tested environmental attribute label type (n = 19), followed by label claims focussing on some aspect of ecological impact (n = 10). Water and carbon-footprint schemes were not as well represented, tested just one and seven times respectively. Animal welfare was the most frequently tested type of label within the social responsibility attribute area (n = 8). Overall, nutrition-health claims were tested in 14 studies, with calorie and nutrient content information tested in 11 studies. A sub-set of macronutrient labelling focussing on fat content appeared in five studies, with eight studies examining vitamin-specific claims.

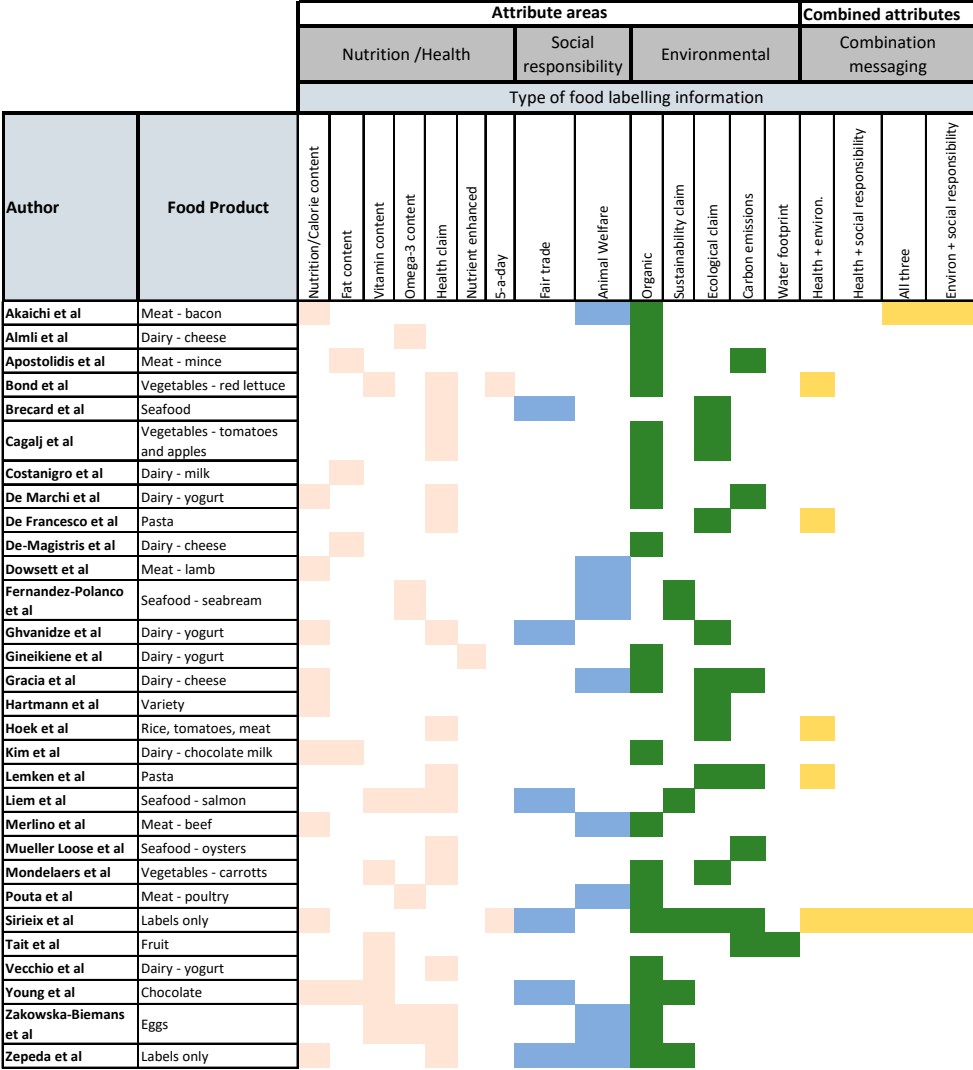

**Figure 4.** Mapping the attributes, food types and labels tested.

### 3.4.3. Foods Tested

Certain food products were tested more frequently than others, with dairy products featuring in nine studies. Meat also featured prominently (n = 6). A pattern can be seen in the relationship between food type and attribute preference with the majority of studies testing dairy finding a preference for organic (six of nine). In contrast, of the ten studies testing meat and fish, four found a preference for animal welfare claims, three for nutrition information and two for combination messaging. Of those foods that can be viewed as 'luxury', non-essential items (oysters, seabream, chocolate, smoked salmon

and chocolate milk), information pertaining to nutrition and ingredient quality was preferred to sustainability credentials (four out of five studies).

### 3.4.4. Combination Labelling

Five studies examined the effect of combining nutrition and/or environmental attribute information (two choice experiments, two experimental auctions and one qualitative study) [40,47,49,50,63]. Four of the five showed a significant increase in both consumer preference and WTP as a result. Only the qualitative study (deemed 'low quality' in the quality assessment) found that nutrition labels were negatively received in combination with social and environmental labels [40].

### 3.4.5. Willingness-to-Pay for Attribute Information

It proved challenging to quantitatively assess WTP across studies, since most used different products, with different absolute prices in various currencies. Therefore, we were able to compare relative WTP percentage price premiums in only seven out of the 12 studies that calculated WTP. (Table 3).

WTP as a price premium was consistently low for vitamin label information and generally positive for environmental attributes. The majority of tested attributes elicited a positive WTP, although one study found participants were not willing to pay a premium for either health or environmental attributes, preferring the 'traditional' status quo pasta product [64]. Another negative WTP was also found in a cross-country comparison study [45], with USA participants unwilling to pay a premium for social welfare/equity attributes.

Combination labelling received consistently high WTP. One study calculated WTP for a mix of nutrition-health and environmental attribute information, with this resulting in a WTP premium of 35%, the highest marginal effect within the study [49]. A similar result can be seen in Bond [63], where combining an organic claim with one for vitamin C led to an 18% price premium compared to 3.7% and 7.4% for the respective label claims in isolation.

### 3.4.6. Drivers of Consumer Preference and Liking

Consumer segmentation and/or latent class analysis based on self-reported characteristics and values were calculated in 21 studies, although few variables proved significant predictors of choice. Of those that were significantly associated with consumer preference, outcomes were variable. Organic consumers were less concerned with price, with four studies finding that the least price-conscious consumers were more likely to choose organic and environmental attributes [50,52,61,64]. Two studies noted that a preference for 'naturalness' positively affected choice of sustainable attributes [53,59].

Education was a significant predictor of attribute choice: Four studies found education was positively associated with a preference for environmental and social attributes [28,48,50,65] while two studies found higher education levels were associated with a preference for the nutrition attribute [62,66]. Gender was an important driver of preference, with women more likely to be concerned with animal welfare issues and nutrition, specifically food fat content [28,44,50,62]. Lifestyle values consistently aligned with attribute preference; nine studies found a relationship between self-reported levels of health or environmental consciousness or knowledge and subsequent attribute preference [45,47,49,53,57,59,61,64].

The observed consumer preference heterogeneity suggests that attribute preference varies significantly between individuals and nationalities. Hartmann et al.'s study of four European countries (France, Poland, Sweden and the United Kingdom) found that the French were more likely to pay a premium for sustainability attributes [59], a finding supported by Almli et al.'s comparison of French and Norwegian preferences [54]. In comparison, the UK and North America appear more concerned with health and nutrition attributes than European countries [42,45].

**Table 3.** Willingness-to-pay a percentage price premium (%) in those studies reviewed reporting significant results.

| | Attribute Areas | | | | | | | | | Combined Attributes |
|---|---|---|---|---|---|---|---|---|---|---|
| | Nutrition | | | | | | Environmental | | | Combined |
| | Type of Food Labelling Information | | | | | | | | | |
| **First Author** | **Vitamin Content** | **5-a-day** | **Health Claim A** | **Health Claim B** | **Functional Claim** | **Organic** | **Ecological Claim** | **Carbon Emissions** | **Water Footprint** | **Health & Environment** |
| *Cagalj et al.* | | | | | | 41.9% (apples) 58.7% (tomatoes) | −5.8% | | | |
| *Mondelaers et al.* | 3% | | | | | | −1% | | | |
| *Pouta et al.* | | | | | | | 5% | | | |
| *Tait et al.* | 6% (UK) 8% (Jap) | | | | | | | 39% (UK) 35% (Jap) | 17% (UK) 21% (Jap) | |
| *Vecchio et al.* | | | | | −5% (label) 36% (+info) | 26% (label) 6% (+info) | | | | |
| *Bond et al.* | 7.4% | 4.7% | 38.8% | 24.4% | | 4% | | | | 18% |
| *Lemken et al.* | | | | | | | | | | 35% |

## 4. Discussion

### 4.1. Summary of Key Findings

Despite doubts around attitudes toward sustainability labelling [19,67] this review found a preference for environmental and social responsibility product attributes. In 57% of studies it was found that consumers evaluated environmental and social responsibility labelling information more favourably than nutrition labelling information. Competing attributes did not lead to consumer disengagement but instead heterogenous attribute choice.

The observed preference for organic labelling is notable. With organic foods commonly perceived as the healthier, more 'natural' alternative to conventionally farmed foods [68–70], preference may not have been driven by environmental concerns alone. Consumer perception that environmentally beneficial products confer 'private' health benefits can be seen in several of the reviewed studies. For example, in Hartmann et al., where the palm oil free-from label commanded a higher healthiness perception than gluten and dairy-free labels [59], and in Gineikiene et al. where organic labelled yogurt had a higher perceived health score than the functional yogurt [46]. With both the nutritional and environmental credentials of organic food production questioned despite positive consumer perceptions of organic schemes, it is important that any benefits of organic production are not overstated by marketers given the varied drivers of organic food choice.

Consumer choice preference and attribute evaluation was affected by both food and label type, suggesting these may be limiting factors when promoting the sustainability credentials of certain foods. Consumers responded less favourably to environmental and social attributes for 'luxury' foods, with those foods that would typically command a price premium more likely to result in a preference for nutrition attributes [41,62,71,72]. Given the need for consumption of animal-based produce to decrease in high-income countries in order to for diets to become more sustainable [25] it is interesting that a majority of studies examined animal products (Figure 2). In contrast to dairy products, where organic was the preferred type of attribute information, animal welfare and combination labelling were preferred by consumers in those studies testing meat and fish. Combination labelling certifying that animal welfare standards have been met, while also providing information on nutritional benefits, may therefore prove effective in encouraging uptake of more sustainably produced animal foods while also driving demand for higher standards throughout the food system.

Different labelling schemes within the same attribute area were also diversely received by consumers. In contrast to the popularity of organic labels within the environmental attribute area, carbon label variants were almost without exception poorly received. All cross-country comparison studies found national differences in preference suggesting that socio-cultural beliefs may play a role in determining attribute preference. With the proliferation of food labelling schemes criticised as confusing [73], and given the observed preference heterogeneity, a better understanding of which labels within each of the three attribute areas are preferred by consumers and best-suited to certain foods is required for more effective uptake of sustainable food purchasing behaviour.

Just five studies combining nutrition and social and/or environmental attributes were found, yet four resulted in an uplift in both WTP and consumer preference. This suggests that articulating sustainability as a set of diverse 'omni-' or 'poly-values' encompassing different issues including health, social and environmental values may indeed be an effective way of appealing to different consumer drivers of behaviour change [74,75]. With increasing interest in the use of so-called omni- or meta-labels [76], combination labelling is a policy that should be seriously considered in the context of promoting sustainable diets and appears to be an acceptable intervention.

### 4.2. This Review's Findings in Context

The preference for environmental attribute information and the observed trend towards a higher WTP price premium supports research finding that consumers are willing to pay a positive premium for eco-labelled goods [21,77], with two recent meta-analyses calculating mean percentage premiums of

12.2% and 16.8% for sustainability attributes [20,78]. Where this study differs from existing evidence [19], is in finding that consumer evaluation of environmental and social responsibility product attributes is more positive than previous estimates.

This review postulates that social desirability bias played a role in explaining the preference for environmental attributes. Nutrition labels often highlight the negative aspects of certain foods rather than positively rewarding consumer purchasing decisions. Indeed, the qualitative evidence suggested that some individuals feel coerced by nutrition labels, rejecting them in favour of more positive environmental and social responsibility attribute messaging [40,42]. In Sirieix et al. the nutrition labels evoked a negative response in contrast to Fairtrade and environmental labels; with the nutrition labels viewed as paternalistic, unnecessary and 'taking the pleasure out of food' [40]. Preference for sustainability attributes may therefore be influenced by a 'green halo', where more positive 'marketing-friendly' messages are preferred by consumers, particularly when they align with an individual's values and beliefs [79].

The 'health halo' effect is a type of cognitive bias whereby an inference of health is assumed based on ambiguous information or claims, which may also explain the findings of this review [80]. Several systematic reviews [77,81,82] have identified an organic 'health halo' with consumers viewing organic as a healthier alternative to conventional products, despite a lack of convincing evidence that organic foods are nutritionally superior [83,84]. Additionally, this review's results tally with a recent review on the increasing popularity of the so-called 'clean-label' trend [81]. While no set definition exists for what constitutes a 'clean-label', the trend encompasses organic, natural and free-from, with health concerns cited as a key determinant of liking. However, this link between consumer perceptions of healthy and environmentally friendly products is something that could perhaps be harnessed in the promotion of sustainable diets [85]. This approach may prove beneficial for approaches using a more holistic 'whole of system' approach to unite the disparate strands of sustainable diets into a set of omni-standards [75].

Poor consumer understanding of nutrition information may have led to a lack of engagement with nutrition attributes in the included studies. Indeed, providing more detailed explanation of what functional health claims meant increased consumer preference for health claims over standard nutrition labelling information. Health claims in this review mostly detailed the benefit individuals stand to gain by eating certain foods and were preferred to nutrient content information when both were presented as choice options to consumers. This supports current thinking that 'softer' scientific health claims can positively bias consumer evaluation [86,87], and supports evidence demonstrating the importance of knowledge as a pre-requisite for effective processing of labelling information [88,89]. Certainly, the lack of consumer preference for carbon labelling in this review could be partially explained by the importance of knowledge in driving attribute preference. The carbon labelling literature shows that a lack of understanding around carbon emission reference values can hinder consumer interpretation and use of carbon labels [90–92].

Education, familiarity with health and environmental issues, and gender (female) were positively associated with a preference for all three types of attribute. While this is likely due in part to sample bias towards highly-educated females (with women often responsible for household food purchasing), this is a consistent finding in labelling research where gender, knowledge and education are often associated with the use of nutrition labels [11,22,89,93]. Here, they are also relevant for environmental and social responsibility labelling preference. This relationship between knowledge, education and a preference for both healthful and environmentally friendly products is therefore important to consider when implementing sustainable food system policies that aim to be equitable.

Study design and quality may also have influenced this review's findings. With roughly a third of included studies declaring a potential conflict-of-interest there is a risk this biased results towards the 'softer' or less scientific, more marketing-friendly, attribute information. Several studies offered monetary incentives for participation which is different to real-world shopping environments where other factors such as price and time pressure all play a part in food choice [94]. Indeed, one study

found that participants expressed more positive sentiments towards sustainability credentials in the initial focus group than they did in the subsequent choice experiment where price was valued more highly [41].

*4.3. Strengths and Limitations*

This study builds on Gallastegui et al.'s previous literature review on consumer response to ecolabels dating back to 2002 [23], using a systematic review process in order to minimise reporting bias. To our knowledge our review is the first to directly compare consumer preference for nutrition and environmental or social responsibility product attributes; of growing relevance as food labelling becomes an increasingly global phenomenon. A large number of interdisciplinary studies from 16 different countries were included for final review resulting in a combined sample of thousands. With global awareness of sustainable diets increasing, this review is relevant to current policymaking and social context given that a majority of included studies were published in the past five years.

Study design, quality and sample bias limit the generalisability of findings. One cause for concern is publication delay; 33.3% of studies were published >2 years after initial data collection (two studies recording a gap of ≥5 years) [60,66]. Choice experiments aim to simulate real-world shopping environments and in doing so reduce hypothetical bias [36,95]. Yet they often involve stylized label alternatives not viewed on the physical product and rarely involve monetary transaction; something likely to result in an overestimation of the importance of labelling information [19]. Comparison of consumer food preference using econometric methods as here, and observational studies of shopping behaviour are therefore required. The lack of evidence from real-world settings (where consumers are confronted with a larger array of labels and foods) means that studies may be a more accurate representation of intention than purchasing behaviour.

Heterogeneity of study methodologies meant it was not possible to conduct a meta-analysis. A narrative approach was taken in this review to synthesise the different study methodologies, attribute information, and range of foods assessed. Although this is an appropriate approach given the study differences, this only provides an overview of the literature and no statistically representative results. This is particularly important to note as the populations in included studies were biased towards females and graduates, thus limiting our ability to apply these findings to a wider population and introducing selection bias. All studies reviewed were conducted in high-income countries with the lack of research from low and middle-income countries a concerning research gap (with estimates that over 50% of calories in these countries now come from packaged goods) [96]. Further research investigating low- and middle-income country consumer preferences for different product attributes would therefore be interesting to explore.

## 5. Conclusions

These findings indicate that environmental and social responsibility labelling schemes are of more value to food consumers than previously suggested. Consumers show a marked preference for organic labels in particular. With health often cited as a motivation for purchasing organic, despite a lack of evidence demonstrating organic food's nutritional superiority compared to conventional production methods, the 'health halo' effects of green marketing are powerful. Thus, care should be taken to ensure this does not negatively impact on public health nutrition by leading some consumers to choose environmentally friendly products, even if less nutritious.

This review has also highlighted several gaps in research that require further study. Further research with more representative study populations is required to obtain more generalisable results. The preference for organic and animal welfare labelling schemes over those for carbon emissions suggests there is a need to carefully consider what information is most impactful when promoting sustainable diets. Hence more research assessing the effect of lesser known schemes is needed before they can be widely used (e.g., water footprint labelling). There may be potential for combining attribute information in order to promote more sustainable diets. An added effect was observed when attributes

were combined and appeared well-received by consumers. Omni-labels combining the two therefore merit further investigation.

Understanding the drivers of sustainable food choice remains problematic. While this review found education and individual lifestyle values were associated with attribute preference, socio-demographic characteristics only weakly predicted choice. Better use of behaviour change theories in research should be encouraged. However, care must be taken to ensure that the increasing focus on sustainable diets alongside growing consumer interest does not lead to labelling interventions with an accompanying price premium that would make sustainable diets economically unviable for many. With labelling interventions putting the onus on consumers to choose 'correctly', manufacturers and governments must also accept responsibility and act concurrently to make sustainable and heathy choices the easy and equitable option.

**Supplementary Materials:** The following are available online at http://www.mdpi.com/2071-1050/11/23/6575/s1, File S1: Database search strategies, Table S2: Quality assessment criteria with justification.

**Author Contributions:** Conceptualization, R.C.A.T. and F.H.; methodology, R.C.A.T., F.H., K.A.B., M.Q. and R.G.; validation, R.C.A.T., R.R. and M.Q.; formal analysis, R.C.A.T. and R.R.; investigation, R.C.A.T.; resources, M.Q.; writing—original draft preparation, R.C.A.T.; writing—review and editing, R.C.A.T., F.H., K.A.B., M.Q. and R.G.; visualization, R.C.A.T., F.H. and R.G.; supervision, F.H., K.A.B., R.G. All authors read and approved the final manuscript.

**Funding:** This research forms part of the Sustainable and Healthy Food Systems (SHEFS) programme funded by the Wellcome Trust's *Our Planet, Our Health* programme [grant number: 205200/Z/16/Z].

**Conflicts of Interest:** The authors declare no conflict of interest. The funders had no role in the design of the study; in the collection, analyses, or interpretation of data; in the writing of the manuscript, or in the decision to publish the results.

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
