# Peer review of "Sustainable Diet Dimensions. Comparing Consumer Preference for Nutrition, Environmental and Social Responsibility Food Labelling: A Systematic Review"

_sustainability, doi:10.3390/su11236575_

Round 1

Reviewer 1 Report

The topic of the study is important and interesting. The aim, results and discussion are formulated in a clear way. However, I have got some suggestions/tips for authors.

Keywords

I think that it is better to decrease the number of keywords.

Introduction

Indicate the law aspect referring to the  labelling (for example 2-3 sentences about the Regulation (EU) No 1169/2011 of the EP and of the Council)

Results

There is no Figure 2 in this section; insert it, please.

Strengths and limitations

Indicate the proper source in the first sentence (instead of 2002).

Author Response

Many thanks for your comments which were very useful. In response to your suggestions:

Keywords - I think that it is better to decrease the number of keywords.

Response: I have deleted four keywords that were deemed less relevant to the paper's content (choice experiment, product attribute, green consumerism, health halo), leaving 7 keywords focussed on labelling.

Introduction - Indicate the law aspect referring to the  labelling (for example 2-3 sentences about the Regulation (EU) No 1169/2011 of the EP and of the Council)

Response: A section discussing the current EU regulatory controls around food labelling (1169/2011 and 1924/2006) has been added at line 59.

Results - There is no Figure 2 in this section; insert it, please.

Response: Figure 2 (PRISMA flow diagram) has now been inserted, line 224.

Strengths and limitations - Indicate the proper source in the first sentence (instead of 2002).

Response: line 471, I have amended the first sentence to indicate the source rather than year of publication and provided the associated reference.

Reviewer 2 Report

Dear Authors,

the study is scientifically sound and of high relevance. My recommendation is to accept it in the present form.

Author Response

Dear Reviewer,

Many thanks for your feedback.

As no alterations were recommended, I have not made any revisions based on your feedback.

With thanks and best wishes.

Reviewer 3 Report

The work "Sustainable Diet Dimensions. Comparing Consumer 2 Preference for Nutrition, Environmental and Social Responsibility Food Labelling: a Systematic Review" is of particular interest to a large category of researchers.

We see a very extensive documentation: ”Six databases were systematically searched for studies examining consumer 24 choice/preference/evaluation of nutrition against environmental and/or social responsibility 25 attributes on food labels. Studies were quality assessed against domain-based criteria and reported 26 using PRISMA guidelines. 30 articles with 19,040 participants met inclusion criteria. Study quality 27 was mixed, with samples biased towards highly-educated females. Environmental and social 28 responsibility attributes were preferred to nutrition attributes in 17 studies (11 environmental and 29 six social), compared to nine where nutrition attributes were valued more highly.”.

I appreciate it is well structured.

The method used and the data reflect an obvious scientific character.

Very important: This review therefore 502 recommends care is taken to ensure this does not negatively impact on public health nutrition by 503 leading some consumers to choose environmentally friendly products even if less nutritious.

The conclusions are pertinent.

That is why I am in favor of publishing this article in your magazine.

Eventually, it may be seen by an English teacher (native).

Author Response

Dear reviewer,

Many thanks for your feedback and comments and I am glad to hear you found the paper's conclusions pertinent.

As no specific alterations were suggested in your comments, I have not made any revisions to the paper based on your feedback.

With thanks and best wishes,